# Astrocytes Reduce Store-Operated Ca^2+^ Entry in Microglia under the Conditions of an Inflammatory Stimulus and Muscarinic Receptor Blockade

**DOI:** 10.3390/ph15121521

**Published:** 2022-12-07

**Authors:** Yoo Jin Kim, You Kyoung Shin, Eunhye Seo, Geun Hee Seol

**Affiliations:** 1Department of Basic Nursing Science, College of Nursing, Korea University, Seoul 02841, Republic of Korea; 2BK21 FOUR Program of Transdisciplinary Major in Learning Health Systems, Graduate School, Korea University, Seoul 02841, Republic of Korea

**Keywords:** astrocytes, microglia, linalyl acetate, lipopolysaccharide, scopolamine, store-operated Ca^2+^ entry

## Abstract

Inflammation and loss of cholinergic transmission are involved in neurodegenerative diseases, but possible interactions between them within neurons, astrocytes, and microglia have not yet been investigated. We aimed to compare store-operated Ca^2+^ entry (SOCE) in neurons, astrocytes, and microglia following cholinergic dysfunction in combination with (or without) an inflammatory stimulus and to investigate the effects of linalyl acetate (LA) on this process. We used the SH-SY5Y, U373, and BV2 cell lines related to neurons, astrocytes, and microglia, respectively. Scopolamine or lipopolysaccharide (LPS) was used to antagonize the muscarinic receptors or induce inflammatory responses, respectively. The concentration of intracellular Ca^2+^ was measured using Fura-2 AM. Treatment with scopolamine and LPS significantly increased SOCE in the neuron-like cells and microglia but not in the scopolamine-pretreated astrocytes. LA significantly reduced SOCE in the scopolamine-pretreated neuron-like cells and microglia exposed to LPS, which was partially inhibited by the Na^+^-K^+^ ATPase inhibitor ouabain and the Na^+^/Ca^2+^ exchanger (NCX) inhibitor Ni^2+^. Notably, SOCE was significantly reduced in the LPS plus scopolamine-pretreated cells mixed with astrocytes and microglia, with a two-fold increase in the applied number of astrocytes. LA may be useful in protecting neurons and microglia by reducing elevated SOCE that is induced by inflammatory responses and inhibiting the muscarinic receptors via Na^+^-K^+^ ATPase and the forward mode of NCX. Astrocytes may protect microglia by reducing increased SOCE under the conditions of inflammation and a muscarinic receptor blockade.

## 1. Introduction

Cognitive decline, a common symptom of neurodegenerative diseases, is not an inevitable outcome of brain aging and is instead closely linked to synaptic alterations [1]. Neurodegeneration is a complex process, but the key destructive processes involved in neurodegenerative diseases, such as Alzheimer’s disease (AD) [2], and even in physiological brain aging [3], are neuroinflammation and loss of cholinergic transmission. For instance, the induction of interleukin (IL)-1α, IL-1β, and IL-6 mRNA is greater in glia cultured from old rats than in those from young rats [4]. Muscarinic signaling, measured by fluorometry GTP–Eu binding assays, is also severely suppressed in the prefrontal cortex and hippocampus of aged rats, a finding that is highly associated with cognitive decline [5], and general muscarinic antagonism in the dorsolateral prefrontal cortex “delay cells” significantly reduces the firing and neuronal activity responsible for working memory [6]. Levels of dopamine, a neurotransmitter related to cognitive function, were significantly decreased when scopolamine (0.1 mM) was infused into the somatodendritic region to block muscarinic receptors [7]. Moreover, it was recently shown that acute activation of glial cells by the peripheral injection of lipopolysaccharide (LPS) in mice causes age-dependent biphasic alterations in the electrophysiological properties of cholinergic neurons [8]. In addition, cytosolic Ca^2+^ levels were significantly increased in LPS (1 μg/mL)-treated rat hippocampal neurons [9]. However, the possible interactions of inflammatory responses and cholinergic dysfunction in neurons, astrocytes, and microglia have not yet been investigated.

Calcium ion (Ca^2+^) is an important second messenger that controls synaptic plasticity; thus, subtle Ca^2+^ dysregulation can lead to abnormal consequences related to neurodegeneration [10]. Two well-known types of synaptic plasticity—long-term potentiation and long-term depression—require Ca^2+^ influx via exquisitely regulated control mechanisms, but these mechanisms are impaired with aging, leading to Ca^2+^ overload [1]. Store-operated Ca^2+^ entry (SOCE) is recognized as a major route for Ca^2+^ influx, not only in nonexcitable cells but also in excitable cells, such as neurons [11]. Neuronal SOCE is significantly increased compared with controls in primary hippocampal cultures expressing presenilin 1 protein, in which the ninth exon responsible for familial AD is deleted [12]. In a mouse model of Huntington’s disease, supranormal synaptic neuronal SOCE resulted in a loss of spines from spiny neurons in a striatal medium [13]. The inhibition of TRPC1 (transient receptor potential canonical 1) channel-dependent SOCE improves synaptic stability in mice with HD [14]. Similarly, the inhibition of SOCE-protected PC12 cells against 1-methyl-4-phenylpyridinium-induced injury by decreasing apoptotic cell death suggests that SOCE inhibition may be an ideal target for reducing neuronal injury in patients with Parkinson’s disease [15]. Therefore, identifying new agents that inhibit neuronal SOCE may lead to viable therapeutic strategies for neurodegeneration.

To date, studies on neurodegeneration have largely focused on the neuronal cellular compartment; however, an increasing body of evidence suggests that all cell types in the nervous system, including neurons, astrocytes, and microglia, should be considered owing to their reciprocal connections [16]. Microglia are responsible for immune surveillance functions that contribute to neuronal activity and synaptic plasticity [17]. In addition, astrocytes play critical roles in antioxidant production and neuronal protection in the brain [18]. Moreover, Ca^2+^ dynamics play an essential role in long-term depression, which reflects the interaction between neurotransmitters released by neurons and gliotransmitters released by astrocytes [19]. Therefore, a more comprehensive understanding of neurodegeneration requires a consideration of the interaction of neurons, astrocytes, and microglia.

Linalyl acetate (LA), one of the main constituents of clary sage (*Salvia sclarea* L.) oil [20], is a fragrance ingredient widely used in cosmetic products, such as soaps, lotions, and creams [21]. LA was reported to have beneficial effects in rodent models for diabetes mellitus [22], ulcerative colitis [23], and rheumatoid arthritis [24]. LA has been reported to induce a transient increase in intracellular Ca^2+^ concentration ([Ca^2+^]_i_) in vascular cells [25]. It also significantly increases [Ca^2+^]_i_ in brain endothelial cells and neuroblastoma cells exposed to oxygen-glucose deprivation/reoxygenation [26]. However, no studies have assessed the effects of LA on [Ca^2+^]_i_ in neurons, astrocytes, and microglia in the context of an inflammatory stimulus and cholinergic dysfunction.

Here, we compared SOCE in the SH-SY5Y, U373, and BV2 cell lines related to neurons, astrocytes, and microglia, respectively. The cells were pretreated with scopolamine to block the muscarinic receptors, with or without an LPS for inflammatory stimulus, and were investigated for effects of LA on these processes.

## 2. Results

### 2.1. The Changes in SOCE Induced by LPS under Scopolamine-Pretreated Conditions or by Scopolamine Alone Are Different in SH-SY5Y, U373, and BV2 Cells

We first studied the effect of LPS on SOCE in scopolamine-pretreated and scopolamine-alone-pretreated SH-SY5Y, U373, and BV2 cells. Treatment with scopolamine and LPS significantly increased SOCE in the SH-SY5Y (*p* = 0.001) (Figure 1A,B) and BV2 (*p* = 0.008) cells compared with the corresponding control (CON) group (Figure 1E,F). In the scopolamine-pretreated U373 cells, LPS exposure showed a slight but nonsignificant tendency towards increased SOCE. The largest increase in LPS-induced SOCE relative to the controls was observed in the scopolamine-pretreated SH-SY5Y cells. Scopolamine treatment alone significantly increased SOCE compared with the CON group only in the SH-SY5Y cells (*p* = 0.001) (Figure 1G,H). These results suggest that SOCE in U373 cells is not affected by scopolamine and LPS treatment, whereas the increased SOCE in SH-SY5Y and BV2 cells is largely due to the effects of scopolamine and LPS, respectively.

### 2.2. LA Decreases LPS Exposure-Induced Increases in SOCE in Scopolamine-Pretreated SH-SY5Y and BV2 Cells

The LPS challenge induces oxidative stress [27], and scopolamine contributes to oxidative stress-mediated neurodegeneration [28]. LPS-induced oxidative stress has been shown to activate stromal interaction molecule 1, a component of the SOCE machinery [29], and H_2_O_2_-induced oxidative stress has been shown to stimulate the Ca^2+^ release-activated Ca^2+^ current [30]. Previously, we revealed that LA had an antioxidant effect [31,32]. Based on these findings, we tested the effects of LA on SOCE in scopolamine-pretreated SH-SY5Y and BV2 cells exposed to LPS by comparing these effects with those of the antioxidant vitamin C [33]. The LPS-induced increase of SOCE was significantly reduced by LA in the scopolamine-pretreated SH-SY5Y (*p* < 0.001) (Figure 2A,B) and BV2 (*p* < 0.001) cells (Figure 2E,F). LA had no statistically significant SOCE inhibitory effect in the CON group. Also, this LPS-induced increase in SOCE was inhibited by vitamin C (VitC) in the scopolamine-pretreated SH-SY5Y (*p* < 0.001) (Figure 2A,B) and BV2 (*p* = 0.001) cells (Figure 2E,F), but was not affected as much as LA.

Next, we evaluated the mechanism of the LA-induced decrease in SOCE in SH-SY5Y and BV2 cells using the Na^+^/K^+^-ATPase inhibitor ouabain (1 µM), Na^+^/Ca^2+^ exchanger (NCX) inhibitor Ni^2+^ (100 µM), and L-type Ca^2+^ channel blocker nifedipine (10 µM). In the scopolamine-pretreated SH-SY5Y cells, an LA-induced reduction in LPS-stimulated SOCE was partially inhibited by ouabain (*p* = 0.002) and Ni^2+^ (*p* < 0.001). Under the application of Ni^2+^ and nifedipine, Ni^2+^ alone-induced partially-inhibited SOCE tended to decrease (Figure 2C,D). Similar trends were observed in the scopolamine-pretreated BV2 cells, where an LA-induced reduction in LPS-stimulated SOCE was significantly attenuated by ouabain (*p* < 0.001) and Ni^2+^ (*p* < 0.001). Under the application of Ni^2+^ and nifedipine, Ni^2+^ alone-induced partially-inhibited SOCE tended to decrease (Figure 2G,H). Nifedipine alone did not have a significant effect on an LA-induced reduction in the LPS-stimulated SOCE in the scopolamine-pretreated SH-SY5Y and BV2 cells. These results indicate that the effects of LA may be related to the activation of the Na^+^-K^+^ ATPase and forward mode of NCX in LPS plus scopolamine pretreated SH-SY5Y and BV2 cells.

### 2.3. Doubling the Number of U373 Cells Inhibits the LPS Exposure-Induced Increases in SOCE in Scopolamine-Pretreated SH-SY5Y + U373 + BV2 Mixed Cells

The results in Figure 1 suggested that U373 cells may protect against the increase in SOCE, which is induced by scopolamine and LPS. Therefore, we investigated the effects of doubling the number of U373 cells (a two-fold increase in the applied number of U373 cells) on SOCE in the scopolamine-pretreated SH-SY5Y + U373 + BV2 mixed cells exposed to LPS. Scopolamine treatment alone significantly increased SOCE compared with that seen in the CON group (*p* = 0.026) in the SH-SY5Y + U373 + BV2 mixed cells. Also, LPS exposure significantly increased SOCE in the scopolamine-pretreated SH-SY5Y + U373 + BV2 (*p* < 0.001) cells compared with the corresponding CON group. The largest increase in SOCE compared with the corresponding CON group was observed in the LPS plus scopolamine-pretreated SH-SY5Y + U373 + BV2 mixed cells. This increase in SOCE, induced by scopolamine and LPS, was significantly decreased by VitC (*p* = 0.001), and LA reduced the abnormally increased SOCE (*p* < 0.001) to a level similar to that in the CON group. Notably, doubling the number of U373 cells significantly inhibited SOCE in the LPS plus scopolamine-pretreated SH-SY5Y + U373 + BV2 mixed cells (*p* < 0.001) (Figure 3A–D).

The mechanisms by which the doubling of the number of U373 cells suppresses SOCE induced by the LPS plus scopolamine in SH-SY5Y + U373 + BV2 mixed cells were investigated using 1 µM ouabain, 100 µM Ni^2+^, and 10 µM nifedipine, as described above. In the LPS plus scopolamine-pretreated SH-SY5Y + U373 + BV2 mixed cells, ouabain tended to counteract the decrease in SOCE, induced by doubling the number of U373 cells. Moreover, the decrease in SOCE that was induced by doubling the number of U373 cells was significantly attenuated by Ni^2+^ (*p* = 0.018) and by Ni^2+^ plus nifedipine (*p* = 0.030) but not by nifedipine alone. These results indicate that the effects of doubling the number of U373 cells may be related to the activation of the forward mode of NCX in LPS plus scopolamine-pretreated SH-SY5Y + U373 + BV2 mixed cells (Figure 3E,F).

### 2.4. Doubling the Number of U373 Cells Inhibits the LPS Exposure-Induced Increase in SOCE in the Scopolamine-Pretreated U373 + BV2 Mixed Cells

Finally, we explored the role of U373 cells in the scopolamine-pretreated U373 + BV2 mixed cells with or without LPS exposure. SOCE was not significantly increased in the U373 + BV2 mixed cells treated with scopolamine alone compared with the CON group. However, LPS significantly increased SOCE in the scopolamine-pretreated U373 + BV2 mixed cells compared with the CON group (*p* = 0.002), and this change was significantly inhibited by doubling the number of U373 cells (*p* = 0.014). These results suggest that the doubling of the U373 cells may protect BV2 cells against the increase in SOCE via the forward mode of NCX under conditions of inflammation and a muscarinic receptor blockade (Figure 4A–D).

## 3. Discussion

This study sought to compare SOCE with and without LPS stimulation in neuron-like cells, astrocytes, and microglia, in which the muscarinic receptors were blocked by pretreatment with scopolamine, and assess the effects of LA on this process and evaluate its underlying mechanisms. An inflammatory stimulus in the context of a muscarinic receptor blockade significantly increased [Ca^2+^]_i_ not only in the neuron-like cells but also in microglia. Inflammation is an essential immune response that serves to remove antigens or repair damaged tissue, but it can also produce neurotoxic factors that worsen neurodegenerative mechanisms [34,35]. Ca^2+^ plays an important regulatory role in inflammation and immunity [36]. For instance, the stimulation of microglia with LPS and interferon (IFN)-γ acts through increases in [Ca^2+^]_i_ to promote the release of cytotoxic substances, such as NO [37], and the injection of LPS into the brain of rats increases [Ca^2+^]_i_ in hippocampal neurons and impairs spatial memory, as evidenced by increased time searching for a hidden platform in the Morris water-maze task [38]. Thus, inflammatory responses observed in the presence of a muscarinic receptor blockade suggest the possibility that memory impairment would be induced through increases in [Ca^2+^]_i_ in neurons and microglia.

Muscarinic receptors act through the regulation of [Ca^2+^]_i_ in pyramidal neurons [39,40] to impact long-term potentiation, thereby playing an important role in the information processing involved in learning and memory. Hence, antimuscarinic agents that inhibit muscarinic receptors can cause an imbalance in [Ca^2+^]_i_ homeostasis and impair memory [41,42]. In rats administered with scopolamine, spatial memory was impaired and Ca^2+^ uptake in the mitochondria isolated from the brains of these rats increased compared with that of the controls [43]. In addition, intraperitoneally administered scopolamine induced an increase in reactive oxygen species (ROS) in the rats by increasing Ca^2+^ and decreasing the ATP concentration in the mitochondria isolated from the hippocampus [44]. A previous study showed that scopolamine decreased cholinergic cell functioning by reducing the choline acetyltransferase level [45]. Acetylcholine was detected in cultured human astrocytoma cells [46], and microglia-expressed mRNA for muscarinic receptors [47]. On the basis of these previous studies, in the present study, we compared [Ca^2+^]_i_ in neuron-like cells, astrocytes, and microglia following treatment with scopolamine to block the muscarinic receptors. Under these scopolamine-only conditions, [Ca^2+^]_i_ increased significantly in the neuron-like cells but not in the astrocytes or microglia. This suggests that neurons are more sensitive to Ca^2+^ dyshomeostasis than other cells under muscarinic receptor blockade conditions.

In this study, we assessed the effects of LA on increases in SOCE, with or without an inflammatory stimulus, under muscarinic receptor-blockade conditions and investigated the underlying mechanisms. LA significantly attenuated the increase in LPS-induced SOCE in the neuron-like cells and microglia pretreated with scopolamine, restoring SOCE to normal levels; these effects of LA were greater than those of the well-known antioxidant, VitC [33]. Also, LA significantly reduced the increase in SOCE that was induced by muscarinic receptor blockade alone in neuron-like cells. LA has been shown to promote a Ca^2+^ homeostasis-maintaining effect, preventing cell damage by regulating excessive Ca^2+^ influxes in vascular endothelial cells [20]. Its usefulness has also been demonstrated in controlling inflammatory diseases by regulating the increases in [Ca^2+^]_i_ in human mast cells, induced by stimulation with phorbol myristate acetate or the Ca^2+^ ionophore, A23187 [48]. In contrast, LA significantly increased [Ca^2+^]_i_ in neuroblastoma cells and the brain endothelial cells exposed to oxygen-glucose deprivation/reoxygenation (OGD/R), an effect that appeared to contribute to cell-protective effects by enhancing the activity of cell survival signals [26]. In a rat model of scopolamine-induced dementia, the inhalation of lavender essential oil, in which LA is the main component, showed beneficial effects on spatial memory formation, as evidenced by increased spontaneous alternation behaviors [49]. LPS-induced oxidative stress-activated stromal interaction molecule 1, which is a component of the SOCE machinery in B leukocytes [29], and scopolamine reportedly contributed to oxidative stress-mediated neurodegeneration in mice with amnesia [28]. The antioxidant effects of LA have been demonstrated in many disease models, including models of chronic obstructive pulmonary disease [32] and hypertension-related ischemic injury [31]. Moreover, LA significantly decreased NADPH oxidase 2 expression and ROS generation in OGD/R-treated SH-SY5Y and BV2 cells [26]. Therefore, we speculate that the LA-induced reduction of SOCE may be related to its antioxidant effect (Figure 5).

We found that an LA-induced decrease in LPS-stimulated SOCE in scopolamine-pretreated SH-SY5Y and BV2 cells was not affected by nifedipine. This suggests that the effects of LA in our study may be mediated via inhibiting extracellular Ca^2+^ influx through voltage-dependent calcium channels (VDCC). LA significantly inhibited KCl-induced Ca^2+^ influx in murine synaptosomes [50], which supports our findings.

Also, we showed that LA activated the forward mode of NCX. The NCX plays a neuron-protective role and maintains Na^+^ and Ca^2+^ homeostasis by mediating Ca^2+^ entry into neurons under inflammatory conditions [51]. The escape latency significantly increased in a maze task in the deletion of the NCX gene in mice, and [Ca^2+^]_i_ was significantly higher in the deletion of the NCX gene in the hippocampal neurons than the presence of the NCX gene in hippocampal neurons [52]. Na^+^/K^+^ ATPase can interact with NCX to regulate [Ca^2+^]_i_ in the central nervous system, preventing Ca^2+^ overload [53]. The activation of Na^+^/K^+^ ATPase was found to increase NCX forward mode activity, leading to Ca^2+^ extrusion from the cortical neurons [54]. In addition, a previous report showed that Na^+^/K^+^ ATPase and antioxidant activity were significantly decreased in the cerebral cortex of rats exposed to scopolamine compared with the controls, and [Ca^2+^]_i_ was significantly increased in those neurons pretreated with hydrogen superoxide [55]. These previous studies support the conclusion that Na^+^/K^+^ ATPase and the forward mode of the NCX pathways are involved in the LA-induced inhibition of inflammatory stimulus (LPS)-induced SOCE elevation in neuron-like cells and microglia in the context of a muscarinic receptor blockade.

Of note, there was only a slight tendency towards an increase in [Ca^2+^]_i_ among the LPS plus scopolamine-pretreated astrocytes. Therefore, we aimed to investigate the possible protective effect of astrocytes against an increase in SOCE induced by LPS and scopolamine. A two-fold increase in the applied number of astrocytes significantly inhibited SOCE in the LPS plus scopolamine-treated cells mixed with neuron-like cells, astrocytes, and microglia via the forward mode of the NCX, which was comparable to the effects of LA. The NCX plays an important role in maintaining Na^+^ and Ca^2+^ homeostasis by mediating Ca^2+^ entry into cells under inflammatory conditions [51]. Oxidative stress induces a decrease in plasma membrane Ca^2+^ extrusion systems, including the NCX [56]. Treatment with H_2_O_2_ significantly increased the gene expression levels of antioxidants, including metallothionein-I and -II, in human astrocytes [57]. In addition, the activation of the astrocytic nuclear erythroid factor 2/superoxide dismutase antioxidant pathway in cultured astrocytes prevented cocultured neuronal cell death due to Aβ toxicity [58]. Ornithine cycle-related genes, such as *ASS1* and *ODC1*, which encode the factors responsible for eliminating toxic byproducts, such as ammonia, tended to increase in the Aβ-treated astrocytes [59]. Therefore, these antioxidant and antitoxic properties of astrocytes might activate the forward mode of the NCX, leading to the inhibition of LPS-exposure-induced increases in SOCE of scopolamine-pretreated SH-SY5Y + U373 + BV2 cell mixtures. In support of our results, a previous study showed that the supernatants obtained from LPS-exposed astrocytes decreased neuronal apoptosis and neuronal [Ca^2+^]_i_ levels under an inflammatory stimulus, indicating that the astrocytes had a neuroprotective effect [60].

More importantly, we showed for the first time that astrocytes have protective effects against an increase in SOCE induced by LPS and scopolamine in microglia via a two-fold increase in the applied number of astrocytes in the mixed cells (of astrocytes and microglia). Microglial activation can trigger chronic neurodegeneration through its ability to cause the release of neurotoxic factors, such as ROS [61]. The amyloid β (Aβ)-induced activation of microglia can increase spontaneous Ca^2+^ transients [62], and Aβ can increase [Ca^2+^]_i_ levels and interleukin-1β secretion in cultured mice microglia, leading to microglial activation [63]. Therefore, our findings highlight the beneficial contribution of astrocytes to maintaining [Ca^2+^]_i_ in microglia under the conditions of inflammation and a muscarinic receptor blockade.

## 4. Materials and Methods

### 4.1. Experimental Cell Preparations

SH-SY5Y human neuron-like cells, U373 human astrocyte-like cells, and BV2 murine brain microglia were obtained from American Type Culture collection (Manassas, VA, USA). Cells were maintained in Dulbecco’s Modified Eagle Medium (DMEM) containing 10% fetal bovine serum (FBS) and 1% penicillin/streptomycin at 37 °C in a humidified 5% CO_2_ atmosphere. [Ca^2+^]_i_ was measured in the SH-SY5Y, U373, and BV2 cells, and in a mixture of these three cell types. In the latter experiments, the SH-SY5Y, U373, and BV2 cells were cultured at a 7:2:1 ratio, reflecting the proportion of neurons, astrocytes, and microglia, respectively, in the human central nervous system [64]. Immortalized BV2 cells derived from murine neonatal microglia have been extensively used to investigate microglial function, substituting for primary microglia in several experimental settings [61]. Human-derived microglia are rarely used because it is difficult to obtain primary cultures for transfection; thus, murine microglia are more commonly used [65]. BV2 cells have therefore been frequently used, together with other cell lines of human origin, to explore cell–cell interactions [66,67,68]. Before measuring [Ca^2+^]_i_, we pretreated the cells with 0.1 mM scopolamine for 1 h to sufficiently inhibit the muscarinic receptors and then exposed the cells to 1 μg/mL LPS for 200 s to induce inflammatory responses [7,9]. Adherent cultures were pretreated with scopolamine, whereas the cell suspensions were pretreated with LPS. The SH-SY5Y, U373, and BV2 cells, individually and in the indicated combination, were subsequently incubated with or without an appropriate concentration of LA based on a previous study [26].

### 4.2. Measurement of [Ca^2+^]_i_

[Ca^2+^]_i_ was determined as previously described [25]. Briefly, approximately 1 × 10^6^ cells were loaded with 2.5 µM Fura-2 AM, a cell-permeant acetoxymethyl ester of the Ca^2+^-binding fluorescent dye Fura-2, for 30 min at room temperature. [Ca^2+^]_i_ was measured at an emission wavelength of 510 nm with fluorescence detection at alternating excitation wavelengths of 340 nm and 380 nm using a fluorescence spectrometer (Photon Technology Instruments, Birmingham, NJ, USA). The maximum 340/380 nm fluorescence (R_max_) was measured in the presence of 5 µM ionomycin, a Ca^2+^ ionophore, and minimum 340/380 nm fluorescence (R_min_) was determined after adding 10 mM ethylene glycol-bis(β-aminoethylether)-N,N,N’,N’-tetraacetic acid (EGTA), a Ca^2+^ chelator. The composition of the extracellular fluid was 150 mM NaCl, 10 mM glucose, 10 mM HEPES, 6 mM KCl, 1.5 mM CaCl_2_, and 1 mM MgCl_2_, with the final pH adjusted to pH 7.4 with NaOH. Ca^2+^-free extracellular fluid was prepared by adding 2 mM EGTA.

SOCE was induced by depleting the endoplasmic reticulum (ER) of Ca^2+^ by adding 30 µM 2′,5′-di (tert-butyl)-1,4-benzohydroquinone (BHQ) to the Ca^2+^-free solution, and then adding 1.5 mM Ca^2+^ to activate store-operated Ca^2+^ channels [69].

### 4.3. Solutions and Chemicals

The DMEM used for cell culture was purchased from Welgene (Gyeonsan, South Korea). FBS and penicillin/streptomycin were purchased from Biowest (Riverside, MO, USA), and Fura-2 AM (purity > 95%) was obtained from Invitrogen (Paisley, UK). LA, VitC, scopolamine, EGTA, BHQ, LPS, ionomycin, ouabain, Ni^2+^, nifedipine, and dimethylsulfoxide (DMSO) were obtained from Sigma-Aldrich (St. Louis, MO, USA). Fura-2 AM, BHQ, and LA were dissolved in DMSO, and scopolamine was dissolved in 0.9% saline. All other regents were dissolved in distilled water.

### 4.4. Statistical Analysis

All results are displayed as means ± standard error of the mean (SEM). All statistical analyses were performed using SPSS Statistics version 22 software (IBM, IL, USA). Differences between the groups were analyzed by Student’s *t*-test, and comparisons among multiple groups were analyzed using and one-way analysis of variance followed by least significant deviation posthoc tests, if necessary. Differences were considered statistically significant at *p*-values < 0.05.

## 5. Conclusions

This study is the first to compare SOCE with and without an inflammatory stimulus under muscarinic receptor blockade conditions in neuron-like cells, astrocytes, and microglia to investigate the effects of LA on this process and its underlying mechanisms. Our findings suggest that microglia respond to inflammation, and neuron-like cells respond to muscarinic receptor blockade, with each case causing elevations in SOCE but resulting in differences between the cells. LA decreased SOCE in the neuron-like cells and microglia via the activation of Na^+^/K^+^ ATPase and the forward mode of NCX, which is attributable to inflammatory responses and the muscarinic receptor blockade. A two-fold increase in the applied number of astrocytes attenuated the increase in SOCE caused by an inflammatory stimulus and inhibited the muscarinic receptors in the mixed cells of the astrocytes and microglia. Collectively, these findings suggest that LA may have therapeutic efficacy in neurons and microglia under the conditions of inflammation and muscarinic receptor blockade. Also, the enhancement of astrocytes can be a promising strategy for preventing neurodegenerative diseases caused by inflammation and muscarinic receptor blockade.

## Figures and Tables

**Figure 1 pharmaceuticals-15-01521-f001:**
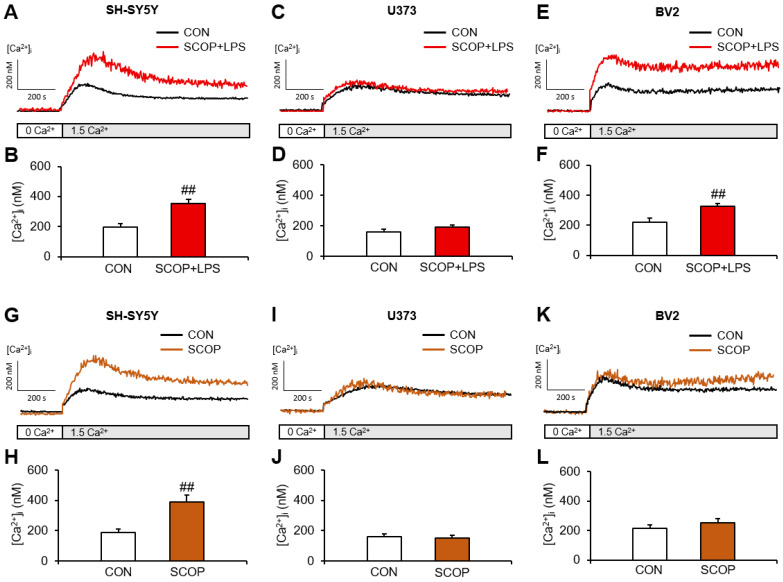
Comparison of store-operated Ca^2+^ entry in SH-SY5Y, U373 and BV2 cells pretreated with scopolamine (SCOP) and treated with lipopolysaccharide (LPS), or treated with SCOP alone. Representative traces showing the effects of LPS in SCOP-pretreated (**A**) SH-SY5Y, (**C**) U373, and (**E**) BV2 cells. Summary data showing the peak [Ca^2+^]_i_ after adding 1.5 mM Ca^2+^ to the (**B**) SH-SY5Y, (**D**) U373, and (**F**) BV2 cells. Representative traces showing effects of SCOP alone to the (**G**) SH-SY5Y, (**I**) U373, and (**K**) BV2 cells. Summary data showing the peak [Ca^2+^]_i_ after adding 1.5 mM Ca^2+^ to the (**H**) SH-SY5Y, (**J**) U373 and (**L**) BV2 cells. Data are reported as means ± SEM (*n* = 6–10; ^##^ *p* < 0.01 vs. the control (CON) group.

**Figure 2 pharmaceuticals-15-01521-f002:**
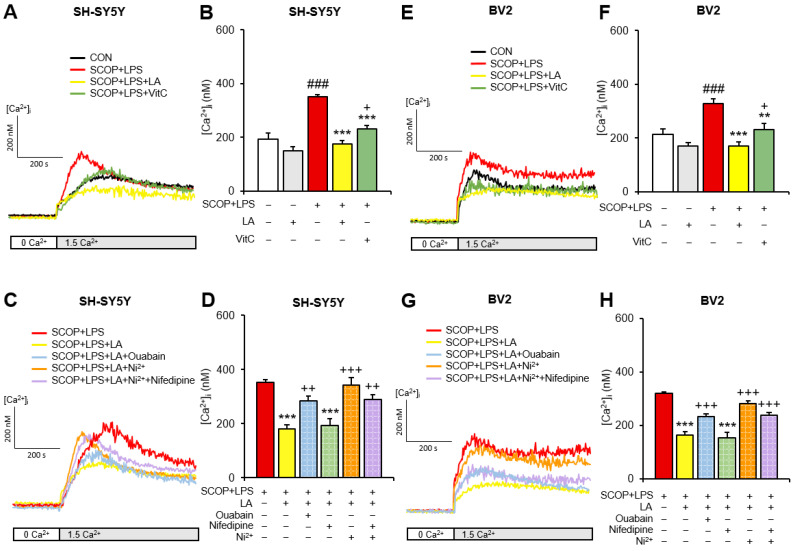
Effects of linalyl acetate (LA) on store-operated Ca^2+^ entry (SOCE) in SH-SY5Y and BV2 cells pretreated with scopolamine (SCOP) and treated with lipopolysaccharide (LPS) and the hypothesized mechanism underlying the LA-induced reduction in SOCE in LPS-exposed, SCOP-pretreated SH-SY5Y and BV2 cells. Representative traces showing the effects of LA on SOCE in scopolamine-pretreated and LPS-exposed (**A**) SH-SY5Y and (**E**) BV2 cells. Summary data showing the peak [Ca^2+^]_i_ observed after the addition of 1.5 mM Ca^2+^ in (**B**) SH-SY5Y and (**F**) BV2 cells. Representative traces showing the effect of ouabain, Ni^2+^, or nifedipine on the LA-induced decreases in SOCE of LPS-exposed, SCOP-pretreated (**C**) SH-SY5Y and (**G**) BV2 cells. Summary data showing the peak [Ca^2+^]_i_ after adding 1.5 mM Ca^2+^ in (**D**) SH-SY5Y and (**H**) BV2 cells. Data are reported as means ± SEM (*n* = 5–9; ^###^ *p* < 0.001 vs. the control (CON) group; ** *p* < 0.01, *** *p* < 0.001 vs. the SCOP+LPS group; ^+^ *p* < 0.05, ^++^ *p* < 0.01, ^+++^ *p* < 0.001 vs. the SCOP + LPS + LA group).

**Figure 3 pharmaceuticals-15-01521-f003:**
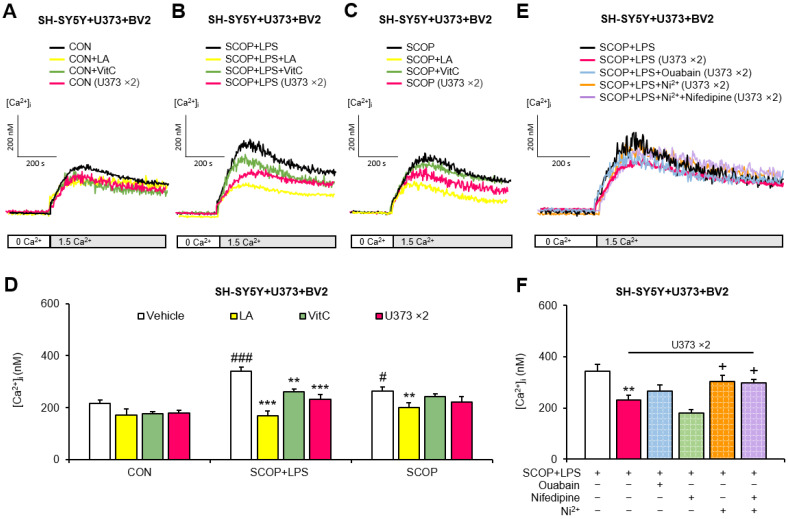
Effects of doubling the number of U373 cells on store-operated Ca^2+^ entry (SOCE) in scopolamine (SCOP)-pretreated SH-SY5Y + U373 + BV2 mixed cells exposed to lipopolysaccharide (LPS), and the hypothesized mechanism by which the doubling in number of U373 cells reduced SOCE in LPS-exposed, SCOP-pretreated SH-SY5Y + U373 + BV2 mixed cells. Representative traces showing the effects of doubling the number of U373 cells on SOCE in the (**A**) control (CON) and (**B**) SCOP+LPS and (**C**) SCOP groups. (**D**) Summary data showing the peak [Ca^2+^]_i_ observed after the addition of 1.5 mM Ca^2+^. (**E**) Representative traces showing the effect of ouabain, Ni^2+^, or nifedipine on the doubled number of U373 cells-induced decreases in SOCE in LPS-exposed, SCOP-pretreated SH-SY5Y + U373 + BV2 mixed cells. (**F**) Summary data showing the peak [Ca^2+^]_i_ observed after the addition of 1.5 mM Ca^2+^. Data are reported as means ± SEM (*n* = 5–9; ^#^
*p* < 0.05, ^###^
*p* < 0.001 vs. the respective CON group; ** *p* < 0.01, *** *p* < 0.001 vs. the respective vehicle group; ^+^
*p* < 0.05 vs. the SCOP+LPS group in doubled U373 cells).

**Figure 4 pharmaceuticals-15-01521-f004:**
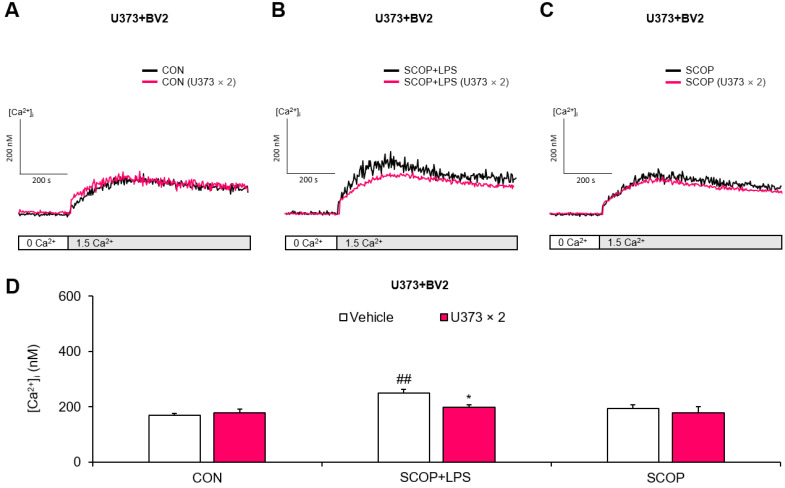
Effects of doubling the number of U373 cells on store-operated Ca^2+^ entry (SOCE) in scopolamine (SCOP)-pretreated U373 + BV2 mixed cells exposed to LPS. Representative traces of the (**A**) control (CON), (**B**) SCOP+LPS, and (**C**) SCOP groups. (**D**) Summary data showing the peak [Ca^2+^]_i_ observed after the addition of 1.5 mM Ca^2+^. Data are reported as means ± SEM (*n* = 5–7; ^##^
*p* < 0.01 vs. the respective CON group; * *p* < 0.05 vs. the respective vehicle group).

**Figure 5 pharmaceuticals-15-01521-f005:**
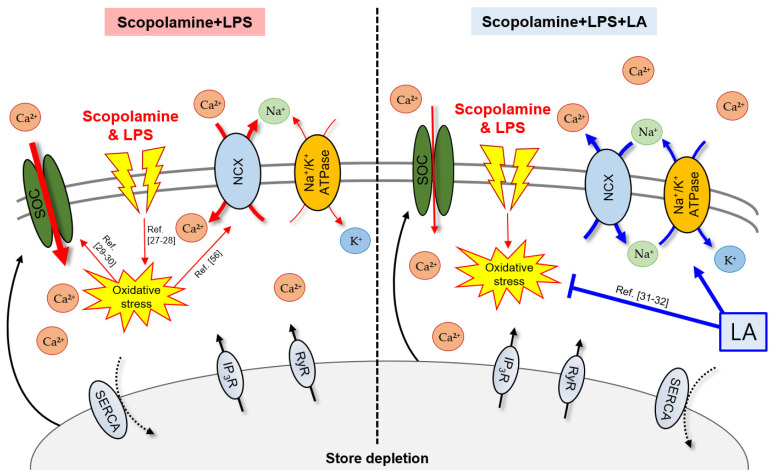
Proposed mechanisms of action of linalyl acetate (LA) on store-operated Ca^2+^ entry (SOCE) in SH-SY5Y and BV2 cells exposed to scopolamine and lipopolysaccharide (LPS). Red arrows reflect the impact of scopolamine and LPS, and blue arrows reflect the effects of LA. The amount of Ca^2+^ influx or efflux is indicated by the thicknesses of the arrows. IP_3_R, inositol 1,4,5-trisphosphate receptor; NCX, Na^+^/Ca^2+^ exchanger; RyR, ryanodine receptor; SERCA, sarcoplasmic reticulum Ca^2+^ ATPase; SOC, store-operated Ca^2+^ channel.

## Data Availability

Data is contained within the article.

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
