# Peer review of "Astrocytes Reduce Store-Operated Ca2+ Entry in Microglia under the Conditions of an Inflammatory Stimulus and Muscarinic Receptor Blockade"

_pharmaceuticals, 2022, doi:10.3390/ph15121521_

Round 1
Reviewer 1 Report
Store-operated calcium entry is a central mechanism in cellular calcium signalling and in maintaining cellular calcium balance. SOCE is the process by which the emptying of endoplasmatic reticulum calcium stores causes influx of calcium ions across the plasma membrane. According to the working hypothesis of the authors the key destructive processes involved in neurodegeneration are neuroinflammation and loss of cholinergic transmission. During their experiments with a biochemical background, the authors aimed to investigate the dynamics of changes in intracellular calcium ion concentrations in three different cell lines in the presence of inflammatory conditions (LPS pretreatment) and muscarinic receptor inhibition (scopolamine pretreatment). In my opinion, the answer to this complex question in the submitted form of the article is not sufficient, so the work needs corrections and additions.
My questions and critical comments follow below.
Compared to the expected complexity of the suggestive term 'downregulation' in the title and objective, it is quite strange that the authors used only one kind of experimental method in this work.
How do the authors justify the inclusion of two human and one mouse cell lines in their experiments?
SH-SY5Y is a neuroblastoma derived cell line for scientific research. Why were primary neuronal cell cultures not used?
Muscarinic receptors are divided into five main subtypes M1, M2, M3, M4, and M5. While each subtype exists within the central nervous system, they are encoded by separate genes and localized to different cell and tissue types. It would be important to know their distribution in neurons, microglial cells and astrocytes.
The experiments presented in the third figure were performed on mixed cell cultures. What was the conceptual reason for studying a 2:1:1 mixture for U373 cells in comparing to the 1:1:1 control suspension containing the same amounts of individual cells?
I do not consider it necessary to provide the calculated ion concentration data and their standard deviation values with an accuracy of two decimal places.
The illustrations in the manuscript are of high quality and show the time course of the change in the intracellular concentration of Ca ions in color. However, it is not clear to me what the bar graphs show or what the 'averaged peak' means. In the explanation of the third figure, it is written that '§§p < 0.01, §§§p < 0.001 vs. the SCOP+LPS group' , but the notation itself is missing from the columns.
I'm not sure if Student’s t-test and ANOVA are appropriate for statistical analysis, since three cell types and multiple treatments are being compared.
It is not clear to the reader whether the LPS and scopolamine pretreatments were performed on adherent cultures or in cell suspensions.
As far as I know, the manuscript has been submitted to the Natural products section of the Journal, so additional information would be necessary regarding the plant (e.g. scientific name) and its active oil-like ingredient studied here, linalyl acetate (LA).
What was the reason for examining LA and only LA in these experiments?
I suggest replacing the term 'downregulation' everywhere in the text with the words ’decrease’ or ’inhibition’ or ’blockade’, and perhaps ’antagonism’ in the case of the muscarinic receptor antagonist scopolamine.
Lines 306-309: These sentences are overlapping and rather meaningless.
In my opinion, the publication of the article requires a significant revision.
Author Response
We would like to thank the reviewers of our manuscript for their valuable comments and helpful suggestions. These comments were insightful and enabled us to improve the quality of our manuscript. The following constitutes a point-by-point response to these comments.
Reviewer 1
Comments and Suggestions for Authors
Store-operated calcium entry is a central mechanism in cellular calcium signalling and in maintaining cellular calcium balance. SOCE is the process by which the emptying of endoplasmatic reticulum calcium stores causes influx of calcium ions across the plasma membrane. According to the working hypothesis of the authors the key destructive processes involved in neurodegeneration are neuroinflammation and loss of cholinergic transmission. During their experiments with a biochemical background, the authors aimed to investigate the dynamics of changes in intracellular calcium ion concentrations in three different cell lines in the presence of inflammatory conditions (LPS pretreatment) and muscarinic receptor inhibition (scopolamine pretreatment). In my opinion, the answer to this complex question in the submitted form of the article is not sufficient, so the work needs corrections and additions.
My questions and critical comments follow below.
Compared to the expected complexity of the suggestive term 'downregulation' in the title and objective, it is quite strange that the authors used only one kind of experimental method in this work.
Response: As suggested by this reviewer, we have changed the word “downregulated” to “reduced”, “decreased” or “inhibited” throughout the manuscript.
How do the authors justify the inclusion of two human and one mouse cell lines in their experiments?
Response: Immortalized BV2 cells derived from murine neonatal microglia have been extensively used to investigate microglial function, substituting for primary microglia in several experimental settings [1]. Human-derived microglia are rarely used because it is difficult to obtain primary cultures for transfection. Thus, murine microglia are more commonly used [2]. BV2 cells have therefore been frequently used, together with other cell lines of human origin, to explore cell-cell interactions [3-5].
We have added this explanation to the Materials and Methods section.
SH-SY5Y is a neuroblastoma derived cell line for scientific research. Why were primary neuronal cell cultures not used?
Response: The human neuroblastoma cell line SH-SY5Y has been used frequently in neurobiological experiments [6]. In addition, most commercially available primary neurons were obtained from neonates; therefore, their behavior may differ from that of adult neurons [7]. The use of primary neurons established from embryonic central nervous system tissue is also limited, because these neurons can no longer be propagated following their terminal differentiation into mature neurons [8]. Therefore, the present study used SH-SY5Y cells rather than primary neurons.
Muscarinic receptors are divided into five main subtypes M1, M2, M3, M4, and M5. While each subtype exists within the central nervous system, they are encoded by separate genes and localized to different cell and tissue types. It would be important to know their distribution in neurons, microglial cells and astrocytes.
Response: Higher levels of M1 than of M2 mRNA, along with detectable levels of M3 mRNA, have been observed in SH-SY5Y cells [9]. These results have led to the use of SH-SY5Y cells to examine muscarinic cholinergic function, especially M1 and M3 [10, 11]. A non-selective M1/M2 muscarinic receptor agonist was found to significantly increase the production of thyrotropin-releasing hormone by U373 cells, suggesting that these cells have functional muscarinic receptors [12]. In addition, BV2 cells were found to express mRNAs encoding M1, M2, M3, M4, and M5 receptors, with the level of M4 mRNA being the highest in these cells [13]. Taken together, these findings showed that SH-SY5Y, U373 and BV2 cells express muscarinic receptors, although the levels of each receptor subtype differed in these cells.
The experiments presented in the third figure were performed on mixed cell cultures. What was the conceptual reason for studying a 2:1:1 mixture for U373 cells in comparing to the 1:1:1 control suspension containing the same amounts of individual cells?
Response: We are sorry for any confusion. In the experiments using a mixture of the three cell types (SH-SY5Y + U373 + BV2), SH-SY5Y, U373, and BV2 cells were cultured in a 7:2:1 ratio, reflecting the in vivo proportions of neurons, astrocytes and microglia, respectively, in the human central nervous system [14]. We have added this detailed information to the Materials and Methods section.
Also, based on the results from Figure 1, we found that store-operated Ca2+ entry (SOCE) into U373 cells was not affected by treatment with scopolamine and LPS, suggesting that U937 cells may protect mixed cultures of SH-SY5Y, U373, and BV2 cells against the increase in SOCE induced by scopolamine and LPS. Therefore, we investigated the effects of doubling the number of U373 cells on SOCE induced by LPS and scopolamine in these mixed cell cultures in order to elucidate whether U373 cells can protect mixed cell cultures against the increase in SOCE induced by scopolamine and LPS.
I do not consider it necessary to provide the calculated ion concentration data and their standard deviation values with an accuracy of two decimal places.
Response: Based on this reviewer’s suggestion, we have deleted the calculated ion concentrations and their standard deviations from the Results section.
The illustrations in the manuscript are of high quality and show the time course of the change in the intracellular concentration of Ca ions in color. However, it is not clear to me what the bar graphs show or what the 'averaged peak' means. In the explanation of the third figure, it is written that '§§p < 0.01, §§§p < 0.001 vs. the SCOP+LPS group', but the notation itself is missing from the columns.
Response: The bar graphs show peak [Ca2+]i observed after the addition of 1.5 mM Ca2+, with the mean values calculated by averaging the data obtained from several repeated experiments. To clarify the meaning of the bar graphs, we have changed the term “averaged peak” to “peak” in all of the figure legends. Also, as this reviewer pointed out, “§§p < 0.01, §§§p < 0.001 vs. the SCOP+LPS group” and “**p < 0.01, ***p < 0.001 vs. the respective vehicle group” overlapped. Therefore, we have deleted “§§p < 0.01, §§§p < 0.001 vs. the SCOP+LPS group” from the legend to Figure 3.
I'm not sure if Student’s t-test and ANOVA are appropriate for statistical analysis, since three cell types and multiple treatments are being compared.
Response: Although there were three cell types and multiple treatments, we mainly focused on comparing the means of independent treatments of the same cell types. For example, in Figure 1B, we compared the SOCE of SH-SY5Y cells in the control group and the SOCE of SH-SY5Y cells treated with scopolamine and LPS. These findings were most appropriately compared using Student’s t-tests and ANOVA.
It is not clear to the reader whether the LPS and scopolamine pretreatments were performed on adherent cultures or in cell suspensions.
Response: In this study, adherent cultures were pretreated with scopolamine, and cell suspensions were pretreated with LPS. We have added this information to the Materials and Methods section.
As far as I know, the manuscript has been submitted to the Natural products section of the Journal, so additional information would be necessary regarding the plant (e.g. scientific name) and its active oil-like ingredient studied here, linalyl acetate (LA).
Response: Linalyl acetate (LA), one of the main constituents of clary sage (Salvia sclarea L.) oil [15], is a fragrance ingredient widely used in cosmetic products, such as soaps, lotions and creams [16]. LA was reported to have beneficial effects in rodent models of diabetes mellitus [17], ulcerative colitis [18] and rheumatoid arthritis [19].
As suggested by this reviewer, we have included the above information in the Introduction section of the manuscript.
What was the reason for examining LA and only LA in these experiments?
Response: We had previously shown that LA induces a transient increase in intracellular Ca2+ concentration ([Ca2+]i), while blocking Ca2+ influx induced by extracellular Ca2+ in vascular cells [20]. LA was also found to significantly increase [Ca2+]i in brain endothelial cells and neuroblastoma cells after oxygen-glucose deprivation/reoxygenation, suggesting that LA protects cells by enhancing the activity of cell survival signals [21]. Moreover, the anti-inflammatory effects of LA [19] suggested that LA could maintain [Ca2+]i in neurons and microglia following inflammatory stimuli and cholinergic dysfunction. Therefore, we evaluated the effects of LA on [Ca2+]i in scopolamine-pretreated neurons and microglia exposed to LPS.
I suggest replacing the term 'downregulation' everywhere in the text with the words ’decrease’ or ’inhibition’ or ’blockade’, and perhaps ’antagonism’ in the case of the muscarinic receptor antagonist scopolamine.
Response: As suggested, we have changed the word “downregulated” to “reduced”, “decreased” or “inhibited” throughout the manuscript.
.
Lines 306-309: These sentences are overlapping and rather meaningless.
Response: We have deleted the first sentence to focus on the beneficial effects of astrocytes in maintaining [Ca2+]i in microglia under conditions of inflammation and muscarinic receptor blockade.
In my opinion, the publication of the article requires a significant revision.
References
- Henn A.; Lund S.; Hedtjärn M.; Schrattenholz A.; Pörzgen P.; Leist M. The suitability of BV2 cells as alternative model system for primary microglia cultures or for animal experiments examining brain inflammation. ALTEX: Alternatives to animal experimentation 2009, 26, 83-94.
- Stansley B.; Post J.; Hensley K. A comparative review of cell culture systems for the study of microglial biology in Alzheimer’s disease. Journal of neuroinflammation 2012, 9, 1-8.
- Lawrimore C.J.; Coleman L.G.; Zou J.; Crews F.T. Ethanol induction of innate immune signals across BV2 microglia and SH-SY5Y neuroblastoma involves induction of IL-4 and IL-13. Brain sciences 2019, 9, 228.
- Liu Y.; Fu Y.; Zhang Y.; Liu F.; Rose G.M.; He X.; Yi X.; Ren R.; Li Y.; Zhang Y. Butein attenuates the cytotoxic effects of LPS-stimulated microglia on the SH-SY5Y neuronal cell line. European Journal of Pharmacology 2020, 868, 172858.
- Pandur E.; Tamási K.; Pap R.; Varga E.; Miseta A.; Sipos K. Fractalkine induces hepcidin expression of BV-2 microglia and causes iron accumulation in SH-SY5Y cells. Cellular and Molecular Neurobiology 2019, 39, 985-1001.
- Feles S.; Overath C.; Reichardt S.; Diegeler S.; Schmitz C.; Kronenberg J.; Baumstark-Khan C.; Hemmersbach R.; Hellweg C.E.; Liemersdorf C. Streamlining Culture Conditions for the Neuroblastoma Cell Line SH-SY5Y: A Prerequisite for Functional Studies. Methods and Protocols 2022, 5, 58.
- Marko D.M.; Foran G.; Vlavcheski F.; Baron D.C.; Hayward G.C.; Baranowski B.J.; Necakov A.; Tsiani E.; MacPherson R.E. Interleukin-6 treatment results in GLUT4 translocation and AMPK phosphorylation in neuronal SH-SY5Y cells. Cells 2020, 9, 1114.
- Kovalevich J.; Langford D.: Considerations for the use of SH-SY5Y neuroblastoma cells in neurobiology. In: Neuronal Cell Culture. Springer; 2013: 9-21.
- Kukkonen J.; Ojala P.; Näsman J.; Hämäläinen H.; Heikkilä J.; Akerman K. Muscarinic receptor subtypes in human neuroblastoma cell lines SH-SY5Y and IMR-32 as determined by receptor binding, Ca++ mobilization and northern blotting. Journal of Pharmacology and Experimental Therapeutics 1992, 263, 1487-1493.
- Eriksson H.; Rössler O.G.; Thiel G. Tyrosine hydroxylase gene promoter activity is upregulated in female catecholaminergic neuroblastoma cells following activation of a Gαq-coupled designer receptor. Neurochemistry International 2022, 160, 105407.
- Naznin F.; Waise T.; Fernyhough P. Antagonism of the muscarinic acetylcholine type 1 receptor enhances mitochondrial membrane potential and expression of respiratory chain components via AMPK in human neuroblastoma SH-SY5Y cells and primary neurons. Molecular Neurobiology 2022, 59, 6754-6770.
- Garcia S.; Porto P.; Martinez V.; Alvarez A.; Finkielman S.; Pirola C. Expression of TRH and TRH-like peptides in a human glioblastoma-astrocytoma cell line (U-373-MG). Journal of endocrinology 2000, 166, 697-703.
- Atwood B.K.; Lopez J.; Wager-Miller J.; Mackie K.; Straiker A. Expression of G protein-coupled receptors and related proteins in HEK293, AtT20, BV2, and N18 cell lines as revealed by microarray analysis. BMC genomics 2011, 12, 1-14.
- Karlsen A.S.; Pakkenberg B. Total numbers of neurons and glial cells in cortex and basal ganglia of aged brains with Down syndrome—a stereological study. Cerebral cortex 2011, 21, 2519-2524.
- Seol G.H.; Lee Y.H.; Kang P.; You J.H.; Park M.; Min S.S. Randomized controlled trial for Salvia sclarea or Lavandula angustifolia: differential effects on blood pressure in female patients with urinary incontinence undergoing urodynamic examination. The journal of alternative and complementary medicine 2013, 19, 664-670.
- Letizia C.; Cocchiara J.; Lalko J.; Api A. Fragrance material review on linalyl acetate. Food and chemical toxicology 2003, 41, 965-976.
- Shin Y.K.; Hsieh Y.S.; Han A.Y.; Kwon S.; Kang P.; Seol G.H. Sex-specific susceptibility to type 2 diabetes mellitus and preventive effect of linalyl acetate. Life Sciences 2020, 260, 118432.
- Shin Y.K.; Kwon S.; Hsieh Y.S.; Han A.Y.; Seol G.H. Linalyl acetate restores colon contractility and blood pressure in repeatedly stressed-ulcerative colitis rats. Environmental Health and Preventive Medicine 2022, 27, 27-27.
- Seo E.; Shin Y.K.; Hsieh Y.S.; Lee J.-M.; Seol G.H. Linalyl acetate as a potential preventive agent against muscle wasting in rheumatoid arthritis rats chronically exposed to nicotine. Journal of Pharmacological Sciences 2021, 147, 27-32.
- You J.H.; Kang P.; Min S.S.; Seol G.H. Bergamot essential oil differentially modulates intracellular Ca2+ levels in vascular endothelial and smooth muscle cells: a new finding seen with fura-2. Journal of cardiovascular pharmacology 2013, 61, 324-328.
- Hsieh Y.S.; Shin Y.K.; Seol G.H. Protection of the neurovascular unit from calcium-related ischemic injury by linalyl acetate. Chinese Journal of Physiology 2021, 64, 88.

Reviewer 2 Report
Kim and colleagues examine SOCE in SH-SY5Y, U373 and BV2 cell lines treated with scopolamine, LPS and linalyl acetate. All experiments are done in the presence of scopolamine to block muscarinic ACh receptors. They find that LPS can induce SOCE in BV2 cells, and that LA can impair SCOP+LPS induced SOCE in SH-SY5Y and BV2. They then show in tri- or bi-cultures of the cell lines that adding twice as many U373 cells can also impair SOCE in the mixed cultures. The experiments appear to be well conducted, but unfortunately I can not recommend the manuscript for publication in Pharmaceuticals. The experiments and rationale are somewhat hard to follow, the importance of the findings is unclear, and some of the conclusions are not sufficiently supported by the data. Please see my specific comments.
1) The rationale for experiments in general is not well explained. The explanations of many experimental details could also be greatly improved.
2) Why have the authors done all experiments in the presence of scopolamine? They indicate that it is to mimic the loss of cholinergic neurons in Alzheimer’s disease, however there is no other clear tie-in to neurodegeneration. The authors do not even demonstrate that each cell line expresses muscarinic receptors.
3) The authors say they are using neurons, astrocytes and microglia, when they are using immortalized cell lines resembling these mature cell types. The three cell lines are not even from the same species (2 human, 1 mouse).
4) The importance of SOCE for many processes remains uncertain, so it is very difficult to appreciate the importance and implications of the findings more broadly.
5) The authors state (line 96-97) “These results suggest that U373 cells may protect against the increase in SOCE induced by LPS and scopolamine” – I do not know what ‘protect against’ means here. A more appropriate conclusion would seem to be that SOCE in U373 cells is not affected by LPS treatment.
6) The authors appear to conclude that LPS increases SOCE in SH-SY5Y cells, though they never examine LPS treatment in the absence of scopolamine. The authors do not compare directly, but it appears the SCOP+LPS effect is no different than SCOP alone. For all of the cell lines the authors should include an LPS only condition (without SCOP).
7) The only clear LPS-induced effect appears to be in BV2 cells, though the authors do not clearly say this.
8) In Fig. 2 the authors show that LA reduces SOCE in SCOP+LPS treated SH-SY5Y and BV2 cells and mention on line 116 as well as in the discussion (line 248) that the effect of Vitamin C is less. They do not however provide a direct statistical comparison of the effects.
9) The authors also do not explain the rationale for testing Vitamin C at all.
10) The authors show in that Ni2+, an inhibitor of NCX, can block prevent the ability of LA to reduce SOCE, however they do not suggest a clear mechanism or model about how blocking the Na/K exchanger could restore SOCE.
11) The authors introduce Fig. 3 by saying “To confirm the possible protective effects of U373 cells…”. Again, I do not think that is an appropriate conclusion from the lack of SOCE observed in U373 cells in Fig. 1.
12) The authors then show that while SOCE is apparently normal (qualitatively similar to in mono-cultures) in co-cultures of SH-SY5Y, U373 and BV2 cells, that adding twice as many U373 cells reduced SOCE. They then show similar effects in co-cultures of U373 and BV2 cells, concluding that U373 cells protect against SCOP+LPS-increased SOCE. Can the authors speculate on a more clear rationale for how U373 cells could be non cell-autonomously affecting SOCE in co-cultured cells (and specifically when there are 2x as many U373 cells, but not when there are 1x U373 cells).
Author Response
We would like to thank the reviewers of our manuscript for their valuable comments and helpful suggestions. These comments were insightful and enabled us to improve the quality of our manuscript. The following constitutes a point-by-point response to these comments.
Reviewer 2
Comments and Suggestions for Authors
Kim and colleagues examine SOCE in SH-SY5Y, U373 and BV2 cell lines treated with scopolamine, LPS and linalyl acetate. All experiments are done in the presence of scopolamine to block muscarinic ACh receptors. They find that LPS can induce SOCE in BV2 cells, and that LA can impair SCOP+LPS induced SOCE in SH-SY5Y and BV2. They then show in tri- or bi-cultures of the cell lines that adding twice as many U373 cells can also impair SOCE in the mixed cultures. The experiments appear to be well conducted, but unfortunately I can not recommend the manuscript for publication in Pharmaceuticals. The experiments and rationale are somewhat hard to follow, the importance of the findings is unclear, and some of the conclusions are not sufficiently supported by the data. Please see my specific comments.
1) The rationale for experiments in general is not well explained. The explanations of many experimental details could also be greatly improved.
Response: In the experiments using a mixture of the three cell types (SH-SY5Y + U373 + BV2), SH-SY5Y, U373, and BV2 cells were cultured in a 7:2:1 ratio, reflecting the in vivo proportions of neurons, astrocytes and microglia, respectively, in the human central nervous system [1]. In addition, adherent cultures were pretreated with scopolamine, whereas cell suspensions were pretreated with LPS. We have added these experimental details to the Materials and Methods section.
2) Why have the authors done all experiments in the presence of scopolamine? They indicate that it is to mimic the loss of cholinergic neurons in Alzheimer’s disease, however there is no other clear tie-in to neurodegeneration. The authors do not even demonstrate that each cell line expresses muscarinic receptors.
Response: Although cholinergic neurons are lost in Alzheimer’s disease, cholinergic deficits have also been associated with neurodegeneration. For example, degeneration of the cholinergic projection system was shown to be associated with the non-motor and motor features of Parkinson’s disease, regardless of dopaminergic denervation [2]. In addition, novel therapeutic interventions have targeted the dysfunctional cholinergic system in Huntington’s disease [3].
Higher levels of M1 than of M2 mRNA, along with detectable levels of M3 mRNA, have been observed in SH-SY5Y cells [4]. These results have led to the use of SH-SY5Y cells to examine muscarinic cholinergic function, especially M1 and M3 [5, 6]. A non-selective M1/M2 muscarinic receptor agonist was found to significantly increase the production of thyrotropin-releasing hormone by the U373 cells, suggesting that these cells have functional muscarinic receptors [7]. In addition, BV2 cells were found to express mRNAs encoding M1, M2, M3, M4, and M5 receptors, with the level of M4 mRNA being highest expression in these cells [8]. Taken together, these findings showed that SH-SY5Y, U373 and BV2 cells express muscarinic receptors, although the levels of each receptor subtype differed in these cells.
3) The authors say they are using neurons, astrocytes and microglia, when they are using immortalized cell lines resembling these mature cell types. The three cell lines are not even from the same species (2 human, 1 mouse).
Response: Immortalized BV2 cells derived from murine neonatal microglia have been extensively used to investigate microglial function, substituting for primary microglia in several experimental settings [9]. Human-derived microglia are rarely used because it is difficult to obtain primary cultures for transfection. Thus, murine microglia are more commonly used [10]. BV2 cells have therefore been frequently used, together with other cell lines of human origin, to explore cell-cell interaction [11-13]. \
We have added this explanation to the Materials and Methods section.
4) The importance of SOCE for many processes remains uncertain, so it is very difficult to appreciate the importance and implications of the findings more broadly.
Response: References on the importance of SOCE in many neurodegenerative diseases have been added to the Introduction section of the manuscript.
5) The authors state (line 96-97) “These results suggest that U373 cells may protect against the increase in SOCE induced by LPS and scopolamine” – I do not know what ‘protect against’ means here. A more appropriate conclusion would seem to be that SOCE in U373 cells is not affected by LPS treatment.
Response: The sentence in the Results suggestion has been changed from “These results suggest that U373 cells may protect against the increase in SOCE induced by LPS and scopolamine.” to “These results suggest that SOCE in U373 cells is not affected by scopolamine and LPS treatment.”
6) The authors appear to conclude that LPS increases SOCE in SH-SY5Y cells, though they never examine LPS treatment in the absence of scopolamine. The authors do not compare directly, but it appears the SCOP+LPS effect is no different than SCOP alone. For all of the cell lines the authors should include an LPS only condition (without SCOP).
Response: We did not examine the effects of LPS alone on SOCE in SH-SY5Y, U373 and BV2 cells. However, based on the results from Figure 1, we speculated on the effects of scopolamine or LPS on SOCE in SH-SY5Y, U373 and BV2 cells. Specifically, treatment of U373 cells with scopolamine or LPS did not significantly alter SOCE when compared with control cells. Treatment with scopolamine alone significantly increased SOCE in SH-SY5Y cells, and treatment with scopolamine and LPS significantly increased SOCE in both SH-SY5Y and BV2 cells. These findings suggested that the increased SOCE in SH-SY5Y cells was largely due to the effects of scopolamine, whereas the increased SOCE in BV2 cells was largely due to the effects of LPS. We have therefore revised the Abstract and Results section accordingly.
7) The only clear LPS-induced effect appears to be in BV2 cells, though the authors do not clearly say this.
Response: See our response to Q6 above. We have revised the Abstract and Results section accordingly.
8) In Fig. 2 the authors show that LA reduces SOCE in SCOP+LPS treated SH-SY5Y and BV2 cells and mention on line 116 as well as in the discussion (line 248) that the effect of Vitamin C is less. They do not however provide a direct statistical comparison of the effects.
Response: We have re-analyzed the differences among groups in Figures 2B and 2F. Although the LPS-induced increase in SOCE was inhibited by vitamin C in scopolamine-pretreated SH-SY5Y cells, it was significantly higher than that of the SCOP+LPS+LA group (p = 0.024). Similarly, SOCE was significantly higher in BV2 cells treated with SCOP+LPS+VitC than with SCOP+LPS+LA (p = 0.029). Accordingly, we have revised Figures 2B and 2F and the Figure legends.
9) The authors also do not explain the rationale for testing Vitamin C at all.
Response: As described in the Discussion section, vitamin C is a well-known antioxidant. LPS-induced oxidative stress was found to activate stromal interaction molecule 1, a component of the SOCE machinery, in B leukocytes [14], and scopolamine was shown to contribute to oxidative stress-mediated neurodegeneration in mice with amnesia [15]. LA has shown antioxidant effects in many disease models, including models of chronic obstructive pulmonary disease [16] and hypertension-related ischemic injury [17]. Therefore, we utilized vitamin C as a positive control, comparing the effects of LA and vitamin C in cells pretreated with scopolamine and LPS.
10) The authors show in that Ni2+, an inhibitor of NCX, can block prevent the ability of LA to reduce SOCE, however they do not suggest a clear mechanism or model about how blocking the Na/K exchanger could restore SOCE.
Response: Na+/K+ ATPase can interact with NCX to regulate [Ca2+]i in the central nervous system, thereby preventing Ca2+ overload [18]. Activation of Na+/K+ ATPase was shown to increase forward mode NCX activity, leading to Ca2+ extrusion by cortical neurons [19]. These findings indicated that LA inhibited SOCE elevation induced by inflammatory responses and inhibited muscarinic receptors via Na+/K+ ATPase.
This explanation has been added to the Discussion section.
11) The authors introduce Fig. 3 by saying “To confirm the possible protective effects of U373 cells…”. Again, I do not think that is an appropriate conclusion from the lack of SOCE observed in U373 cells in Fig. 1.
Response: The results shown in Figure 1 indicate that SOCE in U373 cells was not affected by treatment with scopolamine and LPS. These results suggested that U373 cells protect against the increase in SOCE induced by scopolamine and LPS in mixed cultures of SH-SY5Y+U373+BV2 cells. To determine whether U373 cells could protect against this increase in SOCE, we investigated the effects of doubling the number of U373 cells on SOCE induced by LPS and scopolamine in these mixed cultures.
12) The authors then show that while SOCE is apparently normal (qualitatively similar to in mono-cultures) in co-cultures of SH-SY5Y, U373 and BV2 cells, that adding twice as many U373 cells reduced SOCE. They then show similar effects in co-cultures of U373 and BV2 cells, concluding that U373 cells protect against SCOP+LPS-increased SOCE. Can the authors speculate on a more clear rationale for how U373 cells could be non cell-autonomously affecting SOCE in co-cultured cells (and specifically when there are 2x as many U373 cells, but not when there are 1x U373 cells).
Response: As described above, SH-SY5Y, U373, and BV2 cells were cultured in a 7:2:1 ratio, reflecting the proportions of neurons, astrocytes and microglia, respectively, in the human central nervous system [1]. Therefore, if U373 cells alone could affect SOCE in co-cultured cells, doubling the number of U373 cells would not have reduced SOCE to control levels in LPS plus scopolamine-pretreated SH-SY5Y+U373+BV2 mixed cultures. We found, however, that doubling the number of U373 cells resulted in a significant inhibition of SOCE in these pretreated mixed cell cultures.
References
- Karlsen A.S.; Pakkenberg B. Total numbers of neurons and glial cells in cortex and basal ganglia of aged brains with Down syndrome—a stereological study. Cerebral cortex 2011, 21, 2519-2524.
- Müller M.L.; Bohnen N.I.; Kotagal V.; Scott P.J.; Koeppe R.A.; Frey K.A.; Albin R.L. Clinical markers for identifying cholinergic deficits in Parkinson's disease. Movement Disorders 2015, 30, 269-273.
- D’Souza G.X.; Waldvogel H.J. Targeting the cholinergic system to develop a novel therapy for Huntington’s disease. Journal of Huntington's Disease 2016, 5, 333-342.
- Kukkonen J.; Ojala P.; Näsman J.; Hämäläinen H.; Heikkilä J.; Akerman K. Muscarinic receptor subtypes in human neuroblastoma cell lines SH-SY5Y and IMR-32 as determined by receptor binding, Ca++ mobilization and northern blotting. Journal of Pharmacology and Experimental Therapeutics 1992, 263, 1487-1493.
- Eriksson H.; Rössler O.G.; Thiel G. Tyrosine hydroxylase gene promoter activity is upregulated in female catecholaminergic neuroblastoma cells following activation of a Gαq-coupled designer receptor. Neurochemistry International 2022, 160, 105407.
- Naznin F.; Waise T.; Fernyhough P. Antagonism of the muscarinic acetylcholine type 1 receptor enhances mitochondrial membrane potential and expression of respiratory chain components via AMPK in human neuroblastoma SH-SY5Y cells and primary neurons. Molecular Neurobiology 2022, 59, 6754-6770.
- Garcia S.; Porto P.; Martinez V.; Alvarez A.; Finkielman S.; Pirola C. Expression of TRH and TRH-like peptides in a human glioblastoma-astrocytoma cell line (U-373-MG). Journal of endocrinology 2000, 166, 697-703.
- Atwood B.K.; Lopez J.; Wager-Miller J.; Mackie K.; Straiker A. Expression of G protein-coupled receptors and related proteins in HEK293, AtT20, BV2, and N18 cell lines as revealed by microarray analysis. BMC genomics 2011, 12, 1-14.
- Henn A.; Lund S.; Hedtjärn M.; Schrattenholz A.; Pörzgen P.; Leist M. The suitability of BV2 cells as alternative model system for primary microglia cultures or for animal experiments examining brain inflammation. ALTEX: Alternatives to animal experimentation 2009, 26, 83-94.
- Stansley B.; Post J.; Hensley K. A comparative review of cell culture systems for the study of microglial biology in Alzheimer’s disease. Journal of neuroinflammation 2012, 9, 1-8.
- Lawrimore C.J.; Coleman L.G.; Zou J.; Crews F.T. Ethanol induction of innate immune signals across BV2 microglia and SH-SY5Y neuroblastoma involves induction of IL-4 and IL-13. Brain sciences 2019, 9, 228.
- Liu Y.; Fu Y.; Zhang Y.; Liu F.; Rose G.M.; He X.; Yi X.; Ren R.; Li Y.; Zhang Y. Butein attenuates the cytotoxic effects of LPS-stimulated microglia on the SH-SY5Y neuronal cell line. European Journal of Pharmacology 2020, 868, 172858.
- Pandur E.; Tamási K.; Pap R.; Varga E.; Miseta A.; Sipos K. Fractalkine induces hepcidin expression of BV-2 microglia and causes iron accumulation in SH-SY5Y cells. Cellular and Molecular Neurobiology 2019, 39, 985-1001.
- Hawkins B.J.; Irrinki K.M.; Mallilankaraman K.; Lien Y.-C.; Wang Y.; Bhanumathy C.D.; Subbiah R.; Ritchie M.F.; Soboloff J.; Baba Y. S-glutathionylation activates STIM1 and alters mitochondrial homeostasis. Journal of Cell Biology 2010, 190, 391-405.
- Muhammad T.; Ali T.; Ikram M.; Khan A.; Alam S.I.; Kim M.O. Melatonin rescue oxidative stress-mediated neuroinflammation/neurodegeneration and memory impairment in scopolamine-induced amnesia mice model. Journal of Neuroimmune Pharmacology 2019, 14, 278-294.
- Hsieh Y.S.; Shin Y.K.; Han A.Y.; Kwon S.; Seol G.H. Linalyl acetate prevents three related factors of vascular damage in COPD-like and hypertensive rats. Life sciences 2019, 232, 116608.
- Hsieh Y.S.; Kwon S.; Lee H.S.; Seol G.H. Linalyl acetate prevents hypertension-related ischemic injury. PLoS One 2018, 13, e0198082.
- Sibarov D.A.; Bolshakov A.E.; Abushik P.A.; Krivoi I.I.; Antonov S.M. Na+, K+-ATPase functionally interacts with the plasma membrane Na+, Ca2+ exchanger to prevent Ca2+ overload and neuronal apoptosis in excitotoxic stress. Journal of Pharmacology and Experimental Therapeutics 2012, 343, 596-607.
- Shi M.; Cao L.; Cao X.; Zhu M.; Zhang X.; Wu Z.; Xiong S.; Xie Z.; Yang Y.; Chen J. DR-region of Na+/K+ ATPase is a target to treat excitotoxicity and stroke. Cell Death & Disease 2018, 10, 1-15.

Round 2
Reviewer 1 Report
Since I found the answers to the questions and comments I raised satisfactory and sufficient, I recommend the acceptance of the article.Author Response
We would like to thank the reviewers of our manuscript for their valuable comments and helpful suggestions. These comments were insightful and enabled us to improve the quality of our manuscript. The following constitutes a point-by-point response to these comments.
Reviewer 1
Since I found the answers to the questions and comments I raised satisfactory and sufficient, I recommend the acceptance of the article.
Response: We are very grateful to the reviewer for recommending acceptance of our manuscript. The reviewer’s comments were highly insightful and enabled us to greatly improve the quality of our manuscript. Thank you.

Reviewer 2 Report
The authors have addressed some of my comments, but I still have considerable concerns regarding this manuscript. My specific comments:
1) While the authors have clarified some details, I still find the rationale for design of many experiments, as well as how they are interpreted, difficult to follow. Regarding the latter, I wonder if a model figure could be added to the manuscript that indicates how the authors believe scopolamine, LPS and LA, as well as Vitamin C and the inhibitors are acting mechanistically to affect SOCE? The editors can indicate whether inclusion of such a model figure would be appropriate.
2) Regarding the use of cell lines, I am particularly concerned about SH-SY5Y cells. SH-SY5Y cells possess neuron-like features, but are dividing cells and can not be referred to as 'neurons' unless differentiation is performed. The authors need to be more clear (including in the abstract and introduction) that they are using cell lines related to neurons, astrocytes and microglia. With the SH-SY5Y cells in particular, any reference to them as 'neurons' should be changed to 'neuron-like', 'neuroblastoma', or the cell line name.
3) The authors declined to provide data examining SOCE following LPS only treatment (in the absence of scopolamine). I still believe this should be included in the manuscript, but will leave it up to the editors whether they feel this is necessary.
4) The authors need to provide some sort of explanation/rationale for why they are using vitamin C in the Results section when they are introducing the experiment (section 2.2). The authors do not mention oxidative stress, its potential role in SOCE, or even that LA exhibits anti-oxidant properties in the introduction/results sections. Readers can not be expected to follow the experiments and rationale when this type of background information is not provided.
5) Comment 12 or my initial review asked “Can the authors speculate on a more clear rationale for how U373 cells could be non cell-autonomously affecting SOCE in co-cultured cells (and specifically when there are 2x as many U373 cells, but not when there are 1x U373 cells)”. In response, the authors re-stated their findings without providing any speculation about possible mechanisms that could explain them.
To be more clear, I would like the authors to try to answer: How could having twice as many U373 cells affect SOCE in BV2 cells? What possible cellular mechanism(s) could account for this effect? I would like the authors to discuss possible explanations for their observed results. While they do not need to test it, the authors should be able to provide a plausible explanation for how 2xU373 cells could be reducing SOCE in other cells near them.
Author Response
We would like to thank the reviewers of our manuscript for their valuable comments and helpful suggestions. These comments were insightful and enabled us to improve the quality of our manuscript. The following constitutes a point-by-point response to these comments.
Reviewer 2
Comments and Suggestions for Authors
The authors have addressed some of my comments, but I still have considerable concerns regarding this manuscript. My specific comments:
1) While the authors have clarified some details, I still find the rationale for design of many experiments, as well as how they are interpreted, difficult to follow. Regarding the latter, I wonder if a model figure could be added to the manuscript that indicates how the authors believe scopolamine, LPS and LA, as well as Vitamin C and the inhibitors are acting mechanistically to affect SOCE? The editors can indicate whether inclusion of such a model figure would be appropriate.
Response: Based on the reviewer’s comment, we have added the rationale for the measurement of [Ca2+]i and induction of SOCE in the Materials and Methods. Also, we have added a new figure (Figure 5) outlining the possible mechanisms of action of LA on SOCE in SH-SY5Y and BV2 cells exposed to scopolamine and LPS. The figure is shown below.
Figure 5. Proposed mechanisms of action of linalyl acetate (LA) on store-operated Ca2+ entry (SOCE) in SH-SY5Y and BV2 cells exposed to scopolamine and lipopolysaccharide (LPS). Red arrows reflect the impact of scopolamine and LPS, and blue arrows reflect the effects of LA. The amount of Ca2+ influx or efflux is indicated as the thicknesses of the arrows. IP3R, inositol 1,4,5-trisphosphate receptor; NCX, Na+/Ca2+ exchanger; RyR, ryanodine receptor; SERCA, sarcoplasmic reticulum Ca2+ ATPase; SOC, store-operated Ca2+ channel.
2) Regarding the use of cell lines, I am particularly concerned about SH-SY5Y cells. SH-SY5Y cells possess neuron-like features, but are dividing cells and can not be referred to as 'neurons' unless differentiation is performed. The authors need to be more clear (including in the abstract and introduction) that they are using cell lines related to neurons, astrocytes and microglia. With the SH-SY5Y cells in particular, any reference to them as 'neurons' should be changed to 'neuron-like', 'neuroblastoma', or the cell line name.
Response: We agree with the reviewer’s suggestion. We have added the sentence “We used the SH-SY5Y, U373, and BV2 cell lines related to neurons, astrocytes and microglia, respectively.” in the Abstract and Introduction. Also, based on the reviewer’s comment, we have changed the term “neurons” to “neuron-like cells” throughout the manuscript.
3) The authors declined to provide data examining SOCE following LPS only treatment (in the absence of scopolamine). I still believe this should be included in the manuscript, but will leave it up to the editors whether they feel this is necessary.
Response: As described in the Introduction, loss of cholinergic transmission and neuroinflammation are the main processes involved in neurodegenerative diseases including Alzheimer’s disease [1] and even in physiological brain aging [2]. Therefore, in this study, we primarily focused on investigating SOCE changes induced by scopolamine and LPS in SH-SY5Y, U373 and BV2 cells. Although we did not examine the effects of treatment with LPS alone, we did find that the increased SOCE in SH-SY5Y and BV2 cells was largely due to the effects of scopolamine and LPS, respectively, by comparing the effects of scopolamine plus LPS treatment and scopolamine only treatment.
4) The authors need to provide some sort of explanation/rationale for why they are using vitamin C in the Results section when they are introducing the experiment (section 2.2). The authors do not mention oxidative stress, its potential role in SOCE, or even that LA exhibits anti-oxidant properties in the introduction/results sections. Readers can not be expected to follow the experiments and rationale when this type of background information is not provided.
Response: Thank you for pointing this out. LPS challenge is well known to induce oxidative stress [3], and scopolamine also contributes to oxidative stress-mediated neurodegeneration [4]. LPS-induced oxidative stress has been shown to activate stromal interaction molecule 1, a component of the SOCE machinery [5], and H2O2-induced oxidative stress has been shown to stimulate the Ca2+ release-activated Ca2+ current [6]. Previously we revealed that LA had an antioxidant effect [7, 8]. Based on these findings, we tested the effects of LA on SOCE in scopolamine-pretreated SH-SY5Y and BV2 cells exposed to LPS by comparing these effects with those of the antioxidant, vitamin C [9]. According to the reviewer’s recommendation, we have added this rationale for using vitamin C to the Results section (section 2.2).
5) Comment 12 or my initial review asked “Can the authors speculate on a more clear rationale for how U373 cells could be non cell-autonomously affecting SOCE in co-cultured cells (and specifically when there are 2x as many U373 cells, but not when there are 1x U373 cells)”. In response, the authors re-stated their findings without providing any speculation about possible mechanisms that could explain them. To be more clear, I would like the authors to try to answer: How could having twice as many U373 cells affect SOCE in BV2 cells? What possible cellular mechanism(s) could account for this effect? I would like the authors to discuss possible explanations for their observed results. While they do not need to test it, the authors should be able to provide a plausible explanation for how 2xU373 cells could be reducing SOCE in other cells near them.
Response: We thank the reviewer for this advice. As can be seen in Figure 3F, the effects of doubling the number of U373 cells may be associated with the activation of the forward mode of the NCX in LPS plus scopolamine-pretreated SH-SY5Y+U373+BV2 cell mixtures. The NCX plays an important role in maintaining Na+ and Ca2+ homeostasis by mediating Ca2+ entry into cells under inflammatory conditions [10]. Oxidative stress induces a decrease in plasma membrane Ca2+ extrusion systems, including the NCX [11]. Treatment with H2O2 significantly increased the gene expression levels of anti-oxidants including metallothionein-I and -II in human astrocytes [12]. Also, activation of the astrocytic nuclear erythroid factor 2/superoxide dismutase antioxidant pathway in cultured astrocytes prevented co-cultured neuronal cell death due to Aβ toxicity [13]. Ornithine cycle-related genes, such as ASS1 and ODC1, which encode factors responsible for eliminating toxic byproducts such as ammonia, tended to increase in Aβ-treated astrocytes [14]. Therefore, these anti-oxidant and anti-toxic properties of astrocytes might activate the forward mode of the NCX, leading to inhibition of LPS exposure-induced increases in SOCE of scopolamine-pretreated SH-SY5Y+U373+BV2 cell mixtures. According to the reviewer’s suggestion, we now mention in the Discussion possible cellular mechanisms of astrocytes.
References
- AlFadly E.D.; Elzahhar P.A.; Tramarin A.; Elkazaz S.; Shaltout H.; Abu-Serie M.M.; Janockova J.; Soukup O.; Ghareeb D.A.; El-Yazbi A.F. Tackling neuroinflammation and cholinergic deficit in Alzheimer's disease: Multi-target inhibitors of cholinesterases, cyclooxygenase-2 and 15-lipoxygenase. European journal of medicinal chemistry 2019, 167, 161-186.
- Gamage R.; Wagnon I.; Rossetti I.; Childs R.; Niedermayer G.; Chesworth R.; Gyengesi E. Cholinergic modulation of glial function during aging and chronic neuroinflammation. Frontiers in Cellular Neuroscience 2020, 14, 577912.
- Tao W.; Wang G.; Pei X.; Sun W.; Wang M. Chitosan Oligosaccharide Attenuates Lipopolysaccharide-Induced Intestinal Barrier Dysfunction through Suppressing the Inflammatory Response and Oxidative Stress in Mice. Antioxidants 2022, 11, 1384.
- Muhammad T.; Ali T.; Ikram M.; Khan A.; Alam S.I.; Kim M.O. Melatonin rescue oxidative stress-mediated neuroinflammation/neurodegeneration and memory impairment in scopolamine-induced amnesia mice model. Journal of Neuroimmune Pharmacology 2019, 14, 278-294.
- Hawkins B.J.; Irrinki K.M.; Mallilankaraman K.; Lien Y.-C.; Wang Y.; Bhanumathy C.D.; Subbiah R.; Ritchie M.F.; Soboloff J.; Baba Y. S-glutathionylation activates STIM1 and alters mitochondrial homeostasis. Journal of Cell Biology 2010, 190, 391-405.
- Grupe M.; Myers G.; Penner R.; Fleig A. Activation of store-operated ICRAC by hydrogen peroxide. Cell calcium 2010, 48, 1-9.
- Hsieh Y.S.; Kwon S.; Lee H.S.; Seol G.H. Linalyl acetate prevents hypertension-related ischemic injury. Plos one 2018, 13, e0198082.
- Hsieh Y.S.; Shin Y.K.; Han A.Y.; Kwon S.; Seol G.H. Linalyl acetate prevents three related factors of vascular damage in COPD-like and hypertensive rats. Life sciences 2019, 232, 116608.
- Noh D.; Choi J.G.; Huh E.; Oh M.S. Tectorigenin, a flavonoid-based compound of leopard lily rhizome, attenuates UV-B-induced apoptosis and collagen degradation by inhibiting oxidative stress in human keratinocytes. Nutrients 2018, 10, 1998.
- Fairless R.; Williams S.K.; Diem R. Dysfunction of neuronal calcium signalling in neuroinflammation and neurodegeneration. Cell and tissue research 2014, 357, 455-462.
- Kip S.N.; Strehler E.E. Rapid downregulation of NCX and PMCA in hippocampal neurons following H2O2 oxidative stress. Annals of the New York Academy of Sciences 2007, 1099, 436-439.
- Waller R.; Murphy M.; Garwood C.J.; Jennings L.; Heath P.R.; Chambers A.; Matthews F.E.; Brayne C.; Ince P.G.; Wharton S.B. Metallothionein‐I/II expression associates with the astrocyte DNA damage response and not Alzheimer‐type pathology in the aging brain. Glia 2018, 66, 2316-2323.
- Turati J.; Ramírez D.; Carniglia L.; Saba J.; Caruso C.; Quarleri J.; Durand D.; Lasaga M. Antioxidant and neuroprotective effects of mGlu3 receptor activation on astrocytes aged in vitro. Neurochemistry International 2020, 140, 104837.
- Ju Y.H.; Bhalla M.; Hyeon S.J.; Oh J.E.; Yoo S.; Chae U.; Kwon J.; Koh W.; Lim J.; Park Y.M. Astrocytic urea cycle detoxifies Aβ-derived ammonia while impairing memory in Alzheimer’s disease. Cell Metabolism 2022,

Round 3
Reviewer 2 Report
I now recommend this manuscript for publication.